# Computational Study of Selected Amine and Lactam N-Oxides Including Comparisons of N-O Bond Dissociation Enthalpies with Those of Pyridine N-Oxides

**DOI:** 10.3390/molecules25163703

**Published:** 2020-08-14

**Authors:** Arthur Greenberg, Alexa R. Green, Joel F. Liebman

**Affiliations:** 1Department of Chemistry, University of New Hampshire, Durham, NH 03824, USA; ag1211@wildcats.unh.edu; 2Department of Chemistry and Biochemistry, University of Maryland, Baltimore County, Baltimore, MD 21250, USA

**Keywords:** amine N-oxides, pyridine N-oxide, amide N-oxides, N-O bond dissociation enthalpy

## Abstract

A computational study of the structures and energetics of amine N-oxides, including pyridine N-oxides, trimethylamine N-oxide, bridgehead bicyclic amine N-oxides, and lactam N-oxides, allowed comparisons with published experimental data. Most of the computations employed the B3LYP/6-31G* and M06/6-311G+(d,p) models and comparisons were also made between the results of the HF 6-31G*, B3LYP/6-31G**, B3PW91/6-31G*, B3PW91/6-31G**, and the B3PW91/6-311G+(d,p) models. The range of calculated N-O bond dissociation energies (BDE) (actually enthalpies) was about 40 kcal/mol. Of particular interest was the BDE difference between pyridine N-oxide (PNO) and trimethylamine N-oxide (TMAO). Published thermochemical and computational (HF 6-31G*) data suggest that the BDE of PNO was only about 2 kcal/mol greater than that of TMAO. The higher IR frequency for N-O stretch in PNO and its shorter N-O bond length suggest a greater difference in BDE values, predicted at 10–14 kcal/mol in the present work. Determination of the enthalpy of sublimation of TMAO, or at least the enthalpy of fusion and estimation of the enthalpy of vaporization might solve this dichotomy. The “extra” resonance stabilization in pyridine N-oxide relative to pyridine was consistent with the 10–14 kcal/mol increase in BDE, relative to TMAO, and was about half the “extra” stabilization in phenoxide, relative to phenol or benzene. Comparison of pyridine N-oxide with its acyclic model nitrone (“Dewar-Breslow model”) indicated aromaticity slightly less than that of pyridine.

## 1. Introduction

Amine N-oxides (usually tertiary amines or aromatic amines) are interesting molecules with formally zwitterionic dative N-O bonds that convey high dipole moments and water solubility. Pyridine N-oxides are more reactive as nucleophiles and as electrophiles, compared to the corresponding pyridines [1,2]. Facile oxidation of pyridines to their N-oxides and reduction back to pyridines leads to readily accessible ring substitution reactions than on typically more sluggish pyridines [3,4,5]. The Meisenheimer and Cope reactions utilize the thermal labile nature of the N-O bond, in order to undergo a rearrangement to form an *N*-*N*-disubstituted hydroxylamines [1]. Interestingly, pyridine N-oxide forms a crystalline complex with xenon trioxide that is much less prone to explosion upon thermal or mechanical shock than XeO_3_ itself [6]. Pyridine N-oxide and other heterocyclic N-oxides play significant roles in drug metabolism and as pro-drugs [3,4,7,8]. For example, the pyrrole ring of nicotine is metabolized by flavin-containing monooxygenase 3 (FMO3) in the human liver to (*S*)-nicotine 1′-N-oxide, albeit a relatively minor pathway of nicotine metabolism [7,8,9]. Similarly, morphine is metabolized to its N-oxide in the liver [10]. Trimethylamine is a metabolite derived from choline as well as carnitine. It is oxidized by FMO3 to trimethylamine N-oxide by FMO3 [11]. The 1-azabicyclo[2.2.2]octane (quinuclidine) unit is found in many natural products and this adds to the interest in the quinuclidine N-oxides [12]. Quinuclidine N-oxide carbanion is a weakly nucleophilic strong base, which has found utility in forming carbanions in solution stabilized by the concomitant formation of quinuclidine N-oxide [13].

The nature and strength of the N-O bonds in various amine oxides is of both fundamental and practical interest. The low volatility of these compounds has somewhat limited gas-phase structure studies. Even more limited are the gas-phase enthalpy of formation studies. Aside from the typical challenges in obtaining pure substances, the crystalline amine N-oxides are very hygroscopic, introducing additional challenges in obtaining the extreme accuracy required in the enthalpy of combustion determinations. The low volatilities introduced uncertainties in determinations of enthalpies of sublimation required for standard enthalpies of formation in the gas phase [ΔH°_f_(g) although more formally Δ_f_H°_m_(g)]. The section treating N-O dissociation energies further elaborates on these issues. The dearth of enthalpy data on amine N-oxides is one aspect of the rationale for the current computational study. However, other goals of this study include calculation of N-O bond dissociation enthalpies of bridgehead bicyclic amine N-oxides, including 1-azabicyclo[2.2.2]octane N-oxide (quinuclidine N-oxide) and 1-azabicyclo[3.3.3]undecane N-oxide (manxine N-oxide), which have different strain energies, as well as those of presently unknown corresponding amide N-oxides.

## 2. Results and Discussion

### 2.1. N-O Bond Dissociation Enthalpies

Determination of gas-phase enthalpy data for amine N-oxides is an extremely challenging task [14]. Their formal zwitterionic nature makes them hygroscopic, adding difficulty to accurate combustion experiments. Their low volatility limits the ability to obtain the gas phase data vital to understand the structure and energy of the isolated molecule. The first truly accurate combustion study was said to be that of the crystalline trimethylamine N-oxide published by Steele et al. [14,15]. In their comprehensive review, “The Dissociation Enthalpies of Terminal (N-O) Bonds in Organic Compounds,” Acree et al. estimated the enthalpy of sublimation of this molecule at 80 ± 5 kJ/mol (19.1 ± 1.2 kcal/mol), leading to a gas phase standard enthalpy of formation (ΔH^o^_f_ (CH_3_)_3_NO, g) = −(30 ± 5 kJ/mol) or −7.2 ± 1.2 kcal/mol. Steele also determined the enthalpy of combustion of crystalline pyridine N-oxide [14,15]. In 1988, Shaofeng and Pilcher published a value for crystalline pyridine N-oxide, employing reaction-solution calorimetry, subsequently shown to be too low by 35.5 ± 2.4 kJ/mol (8.5 ± 0.6 kcal/mol) [14,16,17]. Shaofeng and Pilcher also determined the experimental enthalpy of sublimation for pyridine N-oxide [80.6 ± 1.8 kJ/mol (19.3 ± 0.4 kcal/mol)]; this was essentially the value assumed for (CH_3_)_3_NO by Acree et al. Acree et al. agreed to combine Steele’s combustion data with the Shaofeng and Pilcher enthalpy of sublimation, leading to a value of 124.7 ± 1.8 kJ/mol (29.8 ± 0.5 kcal/mol) [14,16,17]. Ribeiro da Silva and her co-workers determined enthalpies of combustion and sublimation for a series of substituted pyridine N-oxides and the corresponding pyridines (as well as the N-oxides of other heterocycles) [17].

Considering the experimental challenges in studying the calorimetry of amine N-oxides, ab initio computational studies offer an opportunity to further explore these molecules with added opportunities to obtain structures and dipole moments among other useful data. An earlier ab initio calculational study (6-31G*) apparently showed good agreement between the hypothetical oxygen atom transfer between pyridine and trimethylamine [18]. Table 1 provides some comparisons of relative enthalpies for molecules useful for calibrating the present study.

A worthwhile comparison is the following isodesmic equation employing the thermochemical data above (pyridine: 33.6 kcal/mol) [19]:(CH_3_)_3_NO + pyridine → (CH_3_)_3_N + pyridine N-oxide ΔH°_r_ = −2.3 kcal/mol(1)

The earlier HF 6-31G* ab initio study provided a calculated energy difference for Equation (1) of −7.1 kJ/mol or −1.7 kcal/mol, in apparently excellent agreement with the reported experimental values [14,18]. The present study reinvestigated this study at the HF 6-31G* and obtained an enthalpy difference of −1.4 kcal/mol, very close to the earlier value. However, as is addressed later, HF 6-31G* predicts a higher dipole moment for pyridine N-oxide, compared to trimethylamine oxide, contrary to intuition and experimental data.

The DFT data in Table 2 predict that Equation (1) is 8–13 kcal/mol more exothermic than that predicted by the HF 6-31G* calculations. Clearly these calculations also differ from the published experimental results, by about the same amount, despite accurately reproducing the experimental thermochemical data listed in Table 1. This discrepancy is analyzed later in this paper.

Employing the experimental for trimethylamine N-oxide (−7.2 kcal/mol) with the corresponding values for trimethylamine (−5.7 kcal/mol) and **^3^**O (59.6 kcal/mol), the experimental bond dissociation energy (BDE) might be obtained for trimethylamine N-oxide (Equation (2)) [19,21]. Employing the experimental values for pyridine N-oxide (+29.8 kcal/mol) and pyridine (+33.6 kcal/mol), the corresponding BDE might be obtained for pyridine N-oxide (Equation (3)).
(CH_3_)_3_NO → (CH_3_)_3_N + ^3^O BDE = 61.1 ± 1.2 kcal/mol(2)
Pyridine N-oxide → pyridine + ^3^O BDE = 63.4 kcal/mol(3)

This is simply another means to display the experimental discrepancy between the five DFT calculations and the experimental data published for Equation (1). It is worth noting an ambitious study employing calculations of TMAO and the three molecules derived by sequentially replacing methyl with phenyl substituents, employing DFT, multireference CASSCF, and the MR-perturbation theory (MCQDPT2) [22]. The N-O BDE value for TMAO was compared with referenced experimental data. However, as argued in the present study, this experimental value had considerable uncertainty, and a better comparison would have been PNO.

Table 3 lists bond dissociation enthalpies as well as Transfer Thermodynamic Reactivity Scale (TTRS) data for a larger group of aliphatic amine N-oxides, pyridine N-oxides and presently-unknown lactam N-oxides. Ribeiro da Silva et al. and Acree et al. included discussions of TTRS [14,17]. It is understood that neither Equations (2) and (3) nor the corresponding dissociations to triplet oxygen listed in Table 2 and Table 3 were isogyric. Thus, caution must be exercised in the interpretation of the computational results. However, comparisons in Table 3 of experimental and computational data for CO_2_ and NO_2_, the first being a non-isogyric reaction, were in quite good agreement at the B3LYP/6-31G* level with significant discrepancies for CO_2_ at the M06/6-311G+(d,p) level, particularly for loss of 1/2 ^3^O_2_. It appears that the major discrepancies between experiment and computation for the N-oxides arose from the BDE and −TTRS of trimethylamine N-oxide. Calculations of the TTRS scale were also not isogyric. Table 3 also includes BDE and TTRS data for two partially-hydrogenated derivatives of pyridine N-oxide, as well as values for carbon dioxide and nitrogen dioxide for the sake of comparison.

It was of interest to compare computed N-O BDE values for some of the substituted pyridine N-oxides that were experimentally determined by Ribeiro da Silva et al. [17]. Although the range of experimental values was only about 5 kcal/mol, they were reasonably well reproduced, computationally. The highest experimental value among the pyridine N-oxides (65.9 kcal/mol) belonged to the 2-carboxyl derivative. X-ray data show a clear intramolecular hydrogen bond (six-membered ring) in the N-oxide, while this was somewhat compensated by the hydrogen bond (five-membered ring) in the corresponding pyridine, computation reproducing experimental data [23,24].

Beyond the comparators, CO_2_ and NO_2_, the data in Table 3 separated into essentially four categories. The pyridine N-oxides had N-O BDEs of roughly 60–66 kcal/mol. A bit higher were two partially-reduced derivatives of pyridine N-oxide with N-O BDEs of 66–69 kcal/mol reflecting extra conjugation with the unsaturated, non-aromatic ring system. About 10-plus kcal/mol lower were aliphatic and alicyclic amine oxides as well as two as-yet unknown molecules, 1-azaadamantane-2-one N-oxide and 1-azabicyclo[2.2.2]octan-2-one N-oxide. Both these molecules possessed fully orthogonal amide linkages lacking resonance energy [25,26]. As such, although in a formal connectivity sense they were lactam N-oxides, they were best regarded as α-ketoamines with an unconjugated lone pair on nitrogen. Sliding down another 10 kcal/mol or so were N-oxides of untwisted or mildly twisted lactams, including *N*-methyl-2-pyrrolidinone-*N*-oxide and 1-azabicyclo[3.3.1]nonan-2-one N-oxide. The former delocalized the nitrogen “lone pair” fully with the adjacent carbonyl to produce 18–20 kcal/mol of resonance energy, while the latter had 50–60% of full resonance stabilization [25,26]. An earlier computational study at the HF 6-31G* level explored structures and energies of amide- and lactam N-oxides [27,28]. Neither *N*-ethyl-2-pyrrolidone nor 1-azabicyclo[3.3.1]nonan-2-one reacted at the ambient temperature with dimethyldioxirane in chloroform [29]. However, under the same conditions, 2,4,6-trimethyl-1-azaadamantane-2-one (“Kirby lactam”) reacted immediately. Although the product mixture was complex, the authors suggested immediate formation of the N-oxide and almost immediate subsequent reaction with “adventitious” water to form the corresponding hydrate of the carbonyl group—the N-oxide of the aminoketone hydrate possibly stabilized by intramolecular hydrogen bonding [29].

### 2.2. N-O Bond Lengths and Vibrational Frequencies in Pyridine N-Oxide and Trimethylamine N-Oxide

There was a very significant discrepancy between the enthalpies of the isodesmic Equation (1) with the experimental thermochemical data, and the 6-31G* prediction cited in Reference [12] and the results of the five calculations cited in the present study. More specifically, the experimental N-O bond dissociation enthalpy of pyridine N-oxide was only 2.3 kcal/mol greater than that of trimethylamine N-oxide (Table 2). The calculations in the present study predict that the pyridine N-oxide bond should have a BDE 10.0–13.5 kcal/mol higher than that of trimethylamine N-oxide.

Table 4 lists experimentally-determined N-O bond lengths for these two molecules, as well as the bond lengths calculated in the present study and in the cited 6-31G* study. The N-O bond in trimethylamine N-oxide was roughly 0.09 Angstroms longer than that in pyridine N-oxide.

The N-O vibrational stretch assigned to pyridine N-oxide was a doubled at 1264 cm^−1^ and 1286 cm^−1^ [35]. The N-O vibrational bond stretch assigned to trimethylamine N-oxide by Giguère and Chin was 937 cm^−1^, although they cited earlier values of 947 cm^−1^ and 943 cm^−1^ by other researchers [36]. Using the following equation,
Freq. = 1/2π (k/m_r_)^1/2^(4)
and making the very simplistic assumption of equal reduced masses (r) for the N-O bonds in the two molecules, led to a force constant (k) ratio (PNO/TMAO) of roughly 1.8. It was also true that the different symmetries (C_2v_ and C_3v_) would cause differences in the extent of mixing of the N-O vibrational mode with the rest of the molecule’s vibrations.

Since the pyridine N-oxide bond length was roughly 0.09 Angstrom shorter and its N-O force constant was roughly 1.8 times that of the corresponding values in trimethylamine N-oxide, it raised questions concerning the very small difference (2.3 kcal/mol) in the cited thermochemical data for Equation (1). Although approximations were cited in deriving the enthalpies of sublimation, the errors introduced would seem small, compared to the discrepancy noted above [14]. This issue is further discussed in the Conclusions.

### 2.3. Experimental and Calculated Proton Affinities

Proton affinities of the amines and amine oxides are interesting and relevant for a variety of reasons. In the conversion of amines to amine N-oxides or N-protonated amines, it is worthwhile to compare the structural effects of transitioning from a three-coordinate nitrogen to a four-coordinate nitrogen, with each of these nitrogen atoms bearing a formal positive charge. Comparison of experimental with computational numbers also provides an additional test of the functionals and basis sets employed. Table 5 lists experimental proton affinities (negative of gas-phase enthalpies of protonation).

The results in Table 5 indicate that the B3LYP/6-31G* calculations were in a somewhat better agreement with the experimental PAs than the M06/6-311G+(d,p) calculations with the former typically very close or 1–3 kcal/mol lower and the latter typically 4–7 kcal/mol, with a much larger discrepancy for the [4.3.3] system. The comparison of experimental and calculated proton affinities served another purpose—an independent check on the calculational relationship between the pyridine N-oxide (PNO) and trimethylamine N-oxide (TMAO) described earlier. To the extent that calculations of cations were generally regarded as more reliable than those of anions, and perhaps even zwitterionic species like amine N-oxides, unless very extended and diffuse basis sets were employed, the following comparison was of interest:PA (TMAO) − PA (PNO) = ΔPAΔPA (Exp’t) = +14.3 kcal/molΔPA (B3LYP/6-31G*) = +17.8 kcal/molΔPA [M06/6-311G+(d,p)] = +14.7 kcal/mol(5)

The experimental PA data employed in Equation (5) data avoid the specific ΔH°**_f_**(g) values for TMAO, PNO, and their N-oxides. The purely experimental ΔPA value could be compared with the calculated enthalpy differences. It is interesting that, while there was slightly better agreement between experimental and calculated PAs at the B3LYP/6-31G* level, the M06/6-311G+(d,p) ΔPA value was in better agreement with the experimental value. Still, the ΔPA values at both levels lend additional credibility to the predictions of the calculations.

A significant part of the motivation for the present study was the question of molecular geometries at bridgehead bicyclic N-oxides and their relationship to ease of formation upon oxidation of the corresponding amines (or lactams). Thus, while the sum of the three C-N-C angles around the bridgehead nitrogen in quinuclidine was ca 328° (essentially trigonal pyramidal), the corresponding sum in manxine was ca 356° (essentially planar, as is known experimentally) [38]. While the proton affinity of quinuclidine was ca 1–3 kcal/mol greater than that of manxine (Table 5), its oxygen affinity (i.e., N-O BDE) was ca 6–8.5 kcal/mol greater than that of manxine (Table 3). The sum of the three C-N-C angles in quinuclidine N-oxide was ca 324°—little changed from the amine. According to Bent’s rules, since the O-substituent was electropositive relative to N, the nitrogen orbital directed to oxygen had slightly higher s-character, the three hybrid orbitals directed to carbons had slightly higher p-character and the C-N-C angles were slightly smaller [39]. The sum of the C-N-C angles in protonated quinuclidine was 331.5°, increasing the s-character of the three hybrid orbitals directed toward carbon. Quinuclidine, its N-oxide and its conjugate acid, all comfortably adopt trigonal pyramidal or tetrahedral geometries. In contrast, the manxine skeleton was forced to undergo some stress to assume tetrahedral geometry at nitrogen. For N-oxide, the sum of the C-N-C angles was ca 337°. This reflected the slight squeezing of C-N-C angles to accommodate the comparatively electropositive O-substituent, combined with the increased coordination from three to four. For the conjugate acid of manxine, the C-N-C angles sum was ca 346°. The inductive effect of replacing the nitrogen lone pair with covalently-bound hydrogen moderated the strain on the bicyclo[3.3.3] system. This explains the experimental and calculated similarities in proton affinities (Table 5) and the significant difference in calculated N-O BDE values (Table 3). Perhaps for esthetic pleasure over practicality, it would be interesting to compare the N-O BDEs for the N-oxides of the 2-quinuclidone and 2-manxinone (Table 3)—both molecules unknown, with those of the corresponding amine N-oxides. The N-O BDE of the [3.3.3] system was calculated to be fully 18 kcal lower than that of the corresponding [2.2.2] system. This reflected both the appreciable resonance energy in the [3.3.3] system that was lost upon formation of the N-oxide (no loss of resonance energy in the [2.2.2] system) and the distortion in the manxine system in transitioning to the N-oxide, as discussed above [26]. Table 6 lists the calculated (gas-phase) enthalpies of the reaction of selected amines and lactams, by hydrogen peroxide or dimethyldioxirane (DMDO).

Some interesting observations could be derived from the data in Table 6. The reactivity of the Kirby lactam (1-aza-2-adamantanone derivative) and the lack of reactivity of 1-azabicyclo[3.3.1]nonan-2-one and *N*-ethylpyrrolidinone with DMDO in chloroform at ambient temperature were consistent with the data [29]. As explained in footnotes a and b in Table 6, the B3LYP/6-31G* were 10 kcal/mol closer to experimental data. These data predict the slight exothermicity for the DMDO reaction for the latter two lactams and 11–14 kcal/mol greater exothermicity for the Kirby lactam parent molecule. The H_2_O_2_ calculations actually predict slight endothermicity for the first two lactams and ca 11 kcal/mol exothermicity for the Kirby lactam parent. An interesting point was that among the molecules listed in Table 6, the presently-unknown 2-manxinone was the most resistant to oxidation at nitrogen. This was a combination of the resonance stabilization in the lactam and the resistance, noted earlier, to increasing the coordination of the near-planar nitrogen to four. Another interesting point (footnotes c and d in Table 6) was that employing the best calculated average (−26.7 kcal/mol) for DMDO as “experimental”, indicated that the exothermicities of the DMDO → Acetone and H_2_O_2_ → H_2_O were virtually equal. All calculations described in this study were gas-phase only. Solvent effects were expected to be significant since H_2_O_2_ is typically employed in aqueous media and DMDO is employed in acetone, as generated or solvent-exchanged (e.g., chloroform). Thus, there are differential effects upon the chemical kinetics as well as the thermochemistry.

### 2.4. How Much Stronger Is the N-O Bond in Pyridine N-oxide vs. Trimethylamine N-Oxide?

It is worthwhile to briefly return to the comparison implicit in isodesmic Equation (1). Is the N-O bond in pyridine N-oxide (PNO) only 2.3 kcal/mol stronger than that of the trimethylamine N-oxide? As noted earlier, while there were accepted data for the enthalpy of sublimation of PNO, there were no experimental data for the enthalpy of sublimation, enthalpy of fusion, and enthalpy of vaporization of TMAO. Acree et al. made a reasonable assumption that the enthalpy of sublimation of TMAO was roughly equal to that of PNO (ca 80 kJ/mol or 19.1 kcal/mol) and noted that the resulting enthalpy for Equation (1) was in close agreement with the HF 6-31G* calculations [14]. However, as noted earlier in the present study, the N-O bond length in PNO was roughly 0.09 Å shorter than that of TMAO (e.g., gas-phase electron diffraction; calculations). Furthermore, the N-O stretching frequency in PNO (ca 1270 cm^−1^), relative to that of TMAO (ca 940 cm^−1^) suggests very crudely a ca 1.8 ratio between the force constants. A cogent argument could be made for the approximate equivalence of enthalpies of vaporization of PNO and TMAO. Equation (6) was employed to estimate the enthalpies of vaporization of monosubstituted hydrocarbons RX [42,43].
ΔH_v_(RX) = 1.12 ñ_C_(R) + 0.3n_Q_(R) + 0.71 + b(X)(6)
where ñ_C_ and n_Q_ are the number of nonquaternary and quaternary carbons, respectively, and b(X) is a parameter based upon the substituent X. If NO is taken as the substituent X, then R equals C_5_H_5_ and C_3_H_9_ for PNO and TMAO, respectively. Since X is assumed to be common to the two molecules (although resonance effects differ), PNO is calculated to have a value for ΔH_v_ ca 2.2 kcal/mol higher than that of TMAO. The experimental dipole moments of TMAO and PNO in benzene were found to be 5.02 Debye and 4.24 Debye, respectively [44]. A more recent determination employing microwave spectroscopy found a value for PNO of 4.13 Debye (gas-phase) [45]. Values calculated in the present work (“gas phase”) were as follows—PNO: 3.93 Debye (B3LYP/6-31G*); 4.39 Debye [M06/6-311G+(d,p)]; TMAO: 4.37 (B3LYP/6-31G*); and 4.94 Debye [M06/6-311G+(d,p)] [46]. It was significant that all five DFT calculations employed in the present study obtained dipole moments that were higher for TMAO than the PNO. This was consistent with the experiment and the expectation of delocalization of the negative charge in PNO into the aromatic ring. In contrast, the dipole moments calculated at the HF 6-31G* level, reversed the order (TMAO, 4.90 Debye; PNO, 5.24 Debye). Extra attraction between TMAO molecules in the liquid should make the ΔH_v_ values of TMAO and PNO more nearly equal. However, it should be remembered that the melting points of PNO (66 °C) and TMAO (220–222 °C) indicate significant difference in the enthalpies of fusion of these two molecules. The high melting point in TMAO was due to the denser packing of molecules compared to PNO, and the higher attraction between dipole moments with a very clear superposition of the N-O groups alternating the N-O and the O-N along an axis (footnotes e and f in Table 4). Thus, it appeared that the ΔH_fus_ value for TMAO might be significantly underestimated. Since the ΔHv values were typically greater than ΔH_fus_ for a given substance, we could roughly break the experimental 80.6 kJ/mol (19.3 kcal/mol) value for PNO into ΔH_v_ = 13 kcal/mol and ΔH_fus_ = 6 kcal/mol. If, as argued above, ΔH_v_ remained the same as for PNO, increasing ΔH_fus_ in TMAO to ca 11 kcal/mol would add ca 6.0 kcal/mol to the ΔH_f_°(g) value for TMAO. This would have the practical effect of weakening the N-O BDE in TMAO by a total of 6.0 + 2.3 (see Equation (1)) = 8.3 kcal/mol relative to PNO. This was closer to the calculational difference [especially with the 6-311+(d,p) basis set] and might be more consistent with the differences in bond N-O bond length and N-O vibrational stretching frequencies.

The above argument seemed to be logical and consistent if inexact. Here, however, is the conundrum. As noted in Table 1, the difference in enthalpy between the ΔH°**_f_** (g) published for (CH_3_)_2_NCH_2_OH (−48.6 kcal/mol) and the semi-experimental estimate for its isomer TMAO (−7.2 kcal/mol) was in excellent agreement with the B3LYP/6-31G* as well as the M06/6-311G+(d,p) calculated differences [14]. Yet, the experimental ΔH_r_ for Equation (1), employing −7.2 kcal per mole and +29.8 kcal/mol, respectively, for ΔH°**_f_** (g) of TMAO and PNO, and well-accepted values for trimethylamine and pyridine, was −2.3 kcal/mol, while the corresponding B3LYP/6-31G* and M06/6-311G+(d,p) values were −13.5 kcal/mol and −10.0 kcal/mol, respectively. In the Rogers and Raplejko study (see Table 1), the authors derive a Benson-type group increment for [C-(H)_2_(N)(O)] = −14.98 kcal/mol, which interpolates [C-(H)_2_(N)_2_] = −12.75 kcal/mol and [C-(H)_2_(O)_2_] = −15.43 kcal/mol. Although the experiments involved in obtaining ΔH°**_f_**(g) for (CH_3_)_2_NCH_2_OH were complex and required some approximations, the value appeared to be reasonable. Indeed, Verevkin [47] had independently obtained a value for[C-(H)_2_(N)_2_] = −12.52 kcal/mol and agreed with the value for [C-(H)_2_(O)_2_] employed by Rogers and Raplejko. The experiments involved in deriving ΔH°**_f_** (g) for (CH_3_)_2_NCH_2_OH were complex and the authors did not provide an estimated overall error limit, although Pedley listed it at ±1.0 kcal/mol [19]. Thus, the problem was as follows—addition of ca 6.0 kcal/mol to the assumed enthalpy of sublimation of TMAO increased the apparent discrepancy of the ΔH°**_f_**(g) between TMAO and (CH_3_)_2_NCH_2_OH (Table 1) from ca 0 to 1 kcal/mol to 6 to 7 kcal/mol, even as it reduced the discrepancy in Equation (1) (equivalently the corresponding BDE values), through this quantity.

### 2.5. Resonance Energy in Pyridine N-Oxide

Over twenty-five years ago Wiberg, Nakaji, and Morgan published a very clever study in which they determined the enthalpy of hydrogenation of the trimer of 1-azacyclopentene and the enthalpy of dissociation leading to the enthalpy of formation of the monomer [48]. Experimental data and ab initio calculations were combined to obtain enthalpies of formation of 1-azacyclohexene and 1-aza-1,3-cyclohexadiene, which were employed to conclude that pyridine had a “resonance energy” slightly less than that of benzene, by about 2 kcal/mol [48]. Those authors noted that a very simple way to estimate aromaticity was to observe the enthalpy of hydrogenation of the first step (ΔH_1_) which, in comparing Equations (7) and (8) suggests roughly 3 kcal/mol less aromaticity in pyridine than benzene, fully consistent with their argument. The difference between the enthalpy of hydrogenation of the first step in Equation (7) or (8), and the hydrogenation of cyclohexene (Equation (8), ΔH_3_ = −28.6 kcal) could be considered as one measure of aromaticity (34.0 kcal/mol in benzene; 31.0 kcal/mol in pyridine) [49].

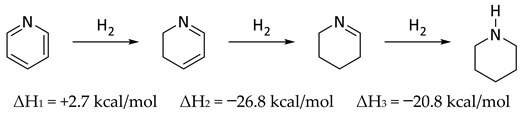
(7)

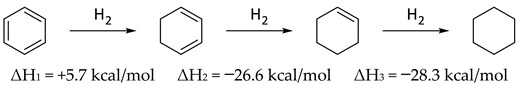
(8)

In the context of the present work, it is interesting to compare the corresponding steps in the reduction of pyridine N-oxide to piperidine N-oxide (Equation (9)). In contrast to Equations (7) and (8), there were no corresponding experimental data for Equation (9) [50]. Indeed, catalytic hydrogenation of pyridine N-oxide produced pyridine [51]. Thus, computational data were employed for Equations (7)–(9) and compared with experimental data, where they existed (Table 7). It is understood that resonance energy in pyridine N-oxide cannot simply be equated to aromaticity, as it was in benzene and pyridine. The oxygen could perhaps be thought of as a substituent on pyridine introducing a zwitterionic character. In this sense there were similarities and significant differences with phenoxide. Based on comparison of ΔH_1_ (pyridine N-oxide) and ΔH_3_ (benzene), the overall calculated resonance energy in pyridine N-oxide was 28.7 kcal/mol (B3LYP/6-31G*) or 34.2 kcal/mol [M06/6-311G+(d,p)].

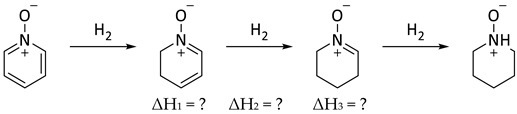
(9)

Although there are many ways to calculate the total resonance energies (RE) of benzene, pyridine, and pyridine N-oxide, perhaps the simplest was offered by isodesmic Equations (10)–(12):RE = 3 ΔH°**_f_** (cyclohexene) − 2 ΔH°**_f_** (cyclohexane) − 2 ΔH°**_f_** (benzene)RE: (exp’t) = 35.6 kcal/mol; (B3LYP/6-31G*) = 39.5 kcal/mol; (M06/6-311G+(d,p) = 37.5 kcal/mol(10)
RE = ΔH°**_f_** (1-azacyclohexene) + 2 ΔH°**_f_** (cyclohexene) − 2 ΔH°**_f_** (cyclohexane) − 2 ΔH°**_f_** (pyridine)RE: (exp’t) = 32.5 kcal/mol; (B3LYP/6-31G*) = 37.5 kcal/mol; (M06/6-311G+(d,p) = 35.7 kcal/mol(11)
RE = ΔH°**_f_** (1-azacyclohexene N-oxide) + 2 ΔH°**_f_** (cyclohexene) − 2 ΔH°**_f_** (cyclohexane) − ΔH_f_^o^ (pyridine N-oxide)RE: (exp’t): n/a; (B3LYP/6-31G*) = 33.7 kcal/mol; [M06/6-311G+(d,p)] = 31.2 kcal/mol.(12)

Comparison of Equations (10)–(12) appeared to show similar total resonance energies for benzene, pyridine, and pyridine N-oxide. However, this comparison was somewhat misleading, in that, in contrast to cyclohexene and 1-azacyclohexene, 1-azacyclohexene N-oxide had considerable resonance stabilization. Thus, it was useful to compare the experimental and calculated enthalpies of saturation for benzene (to cyclohexane) with those of phenol, as well as phenoxide. Table 8 lists those values. The B3PW/6-311G+(d,p) model appears to produce the best agreement with the experiment. Still, what was striking was the increased resonance stabilization using this measure of 10 kcal/mol (M06/6-311G+(d,p) or 14 kcal/mol (B3LYP/6-31G*) of pyridine N-oxide, relative to pyridine, about half the additional stabilization by the substituent in phenoxide. These values were very similar to those calculated to be the increased BDE values in PNO, relative to TMAO (see Table 3). These were about half the enhanced (“extra”) resonance value in phenoxide. In each case, the comparison involved conjugation of an O-substituent on a sp^2^-hybridized atom in an aromatic ring with attachment to a sp^3^-hybridized atom. The difference in the quantity of resonance stabilization arose from phenoxide as anion and pyridine N-oxide as zwitterion. As noted in Table 8, the experimental uncertainty in ΔH°**_f_**(g) (phenoxide) was ca 2.4 kcal/mol. For t-butoxide in Equation (13), the uncertainty was ±12 kJ/mol (ca 2.9 kcal/mol). Therefore, the “extra” resonance energy (phenoxide relative to phenol) in Equation (13) should be taken as 25 ± 6 kcal/mol, essentially the same as the difference in experimental enthalpy of hydrogenation to cyclohexanoxide and cyclohexanol, respectively. The difference was similar for the hydrogenation of benzene. While there was no experimental ΔH°**_f_**(g) value for pyridine N-oxide, the results in Table 8 suggest that the “extra” stabilization in pyridine N-oxide, relative to pyridine was roughly half that in phenoxide, relative to phenol (or cyclohexane).
RE = ΔH°**_f_** (phenol) + ΔH°**_f_** (*t*-butoxide) − ΔH°**_f_** (phenoxide) − ΔH°**_f_** (*t*-butanol)“Extra” RE: (exp’t) = 25.4 kcal/mol(13)

Another interesting approach to evaluating aromaticity is what might be termed the Dewar–Breslow definition [52,53]. Simply put, this approach compared the stability of a conjugated monocyclic polyene with its acyclic analogue (e.g., benzene with 1,3,5-hexatriene; 1,3-cyclobutadiene with 1,3-butadiene; cyclopropenium with allyl cation). While one cannot directly compare ΔH°**_f_**(g) for benzene with that of *E*-1,3,5-hexatriene as they are not isomers, comparison of their heats of hydrogenation (3 mols H_2_ each) allows direct determination of their resonance energy by simply employing Equation (14) (where X = -CH=CH- for benzene and 1,3,5-cyclohexatriene). Hosmane and Liebman noted the experimental limitations in obtaining experimental calorimetric data for N-heterocycles, including pyridine, because of the limited stabilities of the related imines for which there are considerable uncertainties in the very few ΔH°**_f_**(g) values [54]. However, these authors and others noticed that the difference in heats of formation of Ph-X and CH_2_=CH-X was largely independent of X. Furthermore, they noted that 1,2-diphenyl derivatives of imines, for example, are stable and lend themselves to accurate calorimetry. Thus, Equation (15) yields 36.9 kcal/mol for the resonance energy (aromaticity) for benzene (X = -CH=CH-) and 35.2 kcal/mol for pyridine (X = -CH=N-), very similar to the difference published earlier by Wiberg, Nakaji, and Morgan [48,54]. Performing the same analysis (Equation (16)) for pyridine N-oxide [ΔH°**_f_**(g) = +29.8 kcal/mol] and the diphenyl nitrone analogue [ΔH°**_f_**(g) = 62.9 ± 0.5 kcal/mol,], yields a value of 33.1 kcal/mol (ca 2 kcal/mol lower than pyridine) [14]. As noted earlier, pyridine N-oxide, like phenoxide, differs from pyridine and benzene in having a significant π-donor substituent providing “extra resonance energy.” However, the observation that the experimental N-O BDE values in both molecules in Equation (16) were equal (63.3 kcal/mol), suggests the “extra” resonance effects of the O-substituents in pyridine N-oxide and its acyclic model were roughly equal [14]. Thus, the Dewar–Breslow aromaticity values of benzene, pyridine, and pyridine N-oxide were indeed comparable.

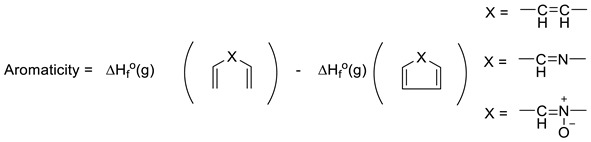
(14)

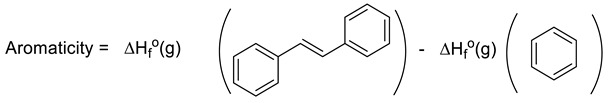
(15)

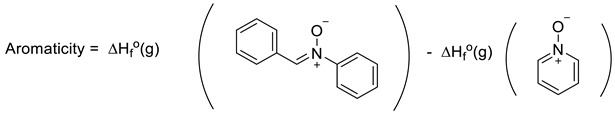
(16)


## 3. Materials and Methods

The present study employed GAUSSIAN 09 [55] and more specifically density functional theory (DFT) and basis sets, as follows. Helpful guidance was provided by the published studies of pyridine N-oxides employing different basis sets and functionals [56,57]. The widely-employed B3LYP/6-31G* DFT/basis set combination was employed in the present study. In addition, the M06/6-311+G(d,p) was paired with all B3LYP/6-31G* calculations, in order to test a more extended and diffuse basis set. Caution needs to be observed in calculating zwitterionic amine oxide compounds due to the unusual nature of the N→O bond, which might even convey some characteristics of anions. The M06 functional was found to give satisfactory results with pyridine N-oxide and was also successful with anions [58]. The B3PW91/6-31G** combination was determined to be optimal for the pyridine N-oxides [57]. The present study also employed B3PW91/6-31G**, as well as B3PW91/6-31G* and B3PW91/6-311G+(d,p) levels for specific test molecules, in order to compare the five levels of calculations. As is demonstrated vide infra the two combinations [B3LYP/6-31G* and M06/6-311+G(d,p)], they generally provide results in good agreement with each other and with the other three levels investigated. All calculations were fully geometry optimized, minima located (all positive frequencies), and the enthalpies reported included zero-point energies and were thermally corrected to 298 K.

## 4. Conclusions

1-Azabicyclo[2.2.2]octane (quinuclidine) and 1-azabicyclo[3.3.3]undecane (manxine) have very similar proton affinities, despite having significantly different geometries (trigonal pyramidal versus near-planar) at the bridgehead nitrogen atoms and significantly different ionization potentials. In contrast, the N-O bond dissociation enthalpy (BDE) was calculated to be 6–8.5 kcal/mol higher (e.g., lower oxygen atom affinity) for manxine than for quinuclidine. This was the result of increased strain in the [3.3.3] system, as its skeleton responded to the attachment of the electropositive O-substituent. Unstrained and lightly-strained amides and lactams have very low oxygen affinities and were not likely to be synthesized, consistent with the previously-reported stabilities of *N*-ethyl-2-pyrrolidinone and 1-azabicyclo[3.3.1]nonan-2-one to dimethyldioxirane, at ambient temperature. In contrast, bridgehead bicyclic lactams, such as 2-quinuclidone and 1-azaadamantane-2-one were calculated to have N-O BDEs comparable to known species, including trimethylamine N-oxide (TMAO). This was consistent with the previously reported reaction of the Kirby lactam with dimethyldioxirane at ambient temperature. The presently-unknown manxine-2-one was calculated to have the lowest BDE, due to the significant resonance energy in the lactam and resistance to accommodating an sp^3^ nitrogen in the [3.3.3] framework. Published data, including an approximation of the enthalpy of sublimation of TMAO, suggest that the BDE of pyridine N-oxide (PNO) was only 2.3 kcal/mol greater than that of TMAO. HF 6-31G* calculations reproduced this BDE difference, but significantly reversed the order in experimentally-observed dipole moments of TMAO and PNO. This small (2.3 kcal/mol) difference in BDE appeared to be inconsistent with the significantly shorter N-O bond in PNO and the significantly higher frequency of the N-O stretch in PNO than in TMAO. The present work postulated that the enthalpy of sublimation of TMAO was underestimated. The conundrum introduced was that, by increasing the enthalpy of sublimation, there was better agreement between with calculated values of the isodesmic Equation (1) (i.e., relative BDE values) but worse agreement between the experimental ΔH°**_f_**(g) between TMAO, its isomer (CH_3_)_2_NCH_2_OH and its calculated values. The DFT calculations employed in the present study suggest that the N-O BDE in PNO was 10–14 kcal/mol higher than that in TMAO. This was consistent with 10–14 kcal/mol “extra resonance energy” in PNO relative to pyridine, which could be compared to 20–25 kcal/mol “extra resonance energy” in phenoxide, relative to benzene or phenol. PNO is a zwitterion, while phenoxide is an anion and greater stabilization through delocalizing a charge over separating opposite charges would be expected. If one employs the Dewar-Breslow approach, the aromaticity of the pyridine N-oxide is about 2 kcal/mol less than that in pyridine, and about 4 kcal/mol less than that in benzene. Very challenging experimental determination of the enthalpy of sublimation (or enthalpy of fusion plus determination or detailed calculation of the enthalpy of vaporization) of TMAO would be a helpful contribution to the study of amine N-oxides. Similarly, enthalpies of hydrogenation of amine N-oxides or enthalpies of oxygen transfer to triphenylphosphine might make valuable contributions to knowledge of N-O BDE values, since enthalpies of reaction are much smaller in magnitude than enthalpies of combustion, and can tolerate larger relative uncertainties. However, a study of oxygen atom transfer (OAT) reactions of molecules including pyridine N-oxide, having N-O BDE values over a 100 kcal/mol range, demonstrated a complex range of mechanisms and complex relationships between the rate of OAT and the N-O BDE values [59].

## Figures and Tables

**Table 1 molecules-25-03703-t001:** Relative experimental and computational differences in standard gas-phase enthalpies of formation for three sets of isomers relevant for the present study. In each column, for each set of isomers, the lowest ΔH°**_f_** (g) was set as 0.0 kcal/mol.

Molecule	Rel. ΔH°_f_ (g) Exp’t	Rel. ΔH°_f_ (g) B3LYP/6-31G*	Rel. ΔH°_f_ (g) M06/6-311G+(d,p)
CH_3_NO_2_	0.0 kcal/mol ^a^	0.0 kcal/mol	0.0 kcal/mol
CH_3_ONO	+2.0 kcal/mol ^a^	+0.9 kcal/mol	+3.3 kcal/mol
(CH_3_)_2_NCH_2_OH	0.0 kcal/mol ^b^	0.0 kcal/mol	0.0 kcal/mol
(CH_3_)_3_NO	+41.4 kcal/mol ^b^	+41.3 kcal/mol	+42.5 kcal/mol
2-HO-pyridine	0.0 kcal/mol ^c^	+1.0 kcal/mol	+0.5 kcal/mol
2-pyridone	+0.7 kcal/mol ^c^	0.0 kcal/mol	0.0 kcal/mol
pyridine N-oxide	+48.9 kcal/mol ^c^	+43.8 kcal/mol	+50.1 kcal/mol

^a^ Pedley: ΔH°**_f_**,g (CH_3_NO_2_) = −17.8 kcal/mol; ΔH°**_f_**,g (CH_3_ONO) = −15.8 kcal/mol) [19]. ^b^ Pedley: ΔH°**_f_**,g ((CH_3_)_2_NCH_2_OH) = −48.6 ± 1.1 kcal/mol [20] (original source: Acree et al. provide ΔH°**_f_**,g [(CH_3_)_3_NO] = −7.2 ± 1.2 kcal/mol [includes assumed enthalpy of sublimation = 80 kJ/mol (19 kcal/mol)] [14,19]. ^c^ Pedley; Lias et al. provide ΔH°**_f_** (g) (2-hydroxypyridine) = 19.0 ± 0.3 kcal/mol and 19 ± 0.5 kcal/mol, respectively; Lias et al. provides ΔH°**_f_** (g) (2-pyridone) = 18 ± 0.5 kcal/mol. For pyridine N-oxide, ΔH°**_f_** (g) = +29.8 kcal/mol [14,17,19,21].

**Table 2 molecules-25-03703-t002:** Calculated energies and enthalpies of reaction (kcal/mol) calculated for the isodesmic Equation (1), as well as bond dissociation energies (BDE) for pyridine N-oxide (PNO) and trimethylamine N-oxide (TMAO) (dissociation to amine + ^3^O) employing six different DFT/Basis Set models.

BDE (PNO) (kcal/mol)	ΔE_r_ (kcal/mol)	ΔH_r_ (kcal/mol)	BDE (PNO) (kcal/mol)	BDE (TMAO) (kcal/mol)
B3LYP/6-31G*	−13.2	−13.5	62.1	48.6
B3LYP-6-31G**	−13.2	−13.5	62.2	48.7
M06/6-311G+(d,p)	−9.7	−10.0	61.5	51.5
B3PW91/6-31G*	−14.3	−14.7	63.8	49.1
B3PW91/6-31G**	−14.4	−14.6	63.9	49.3
B3PW91/6-311G+(d,p)	−11.1	−11.8	62.7	50.9

**Table 3 molecules-25-03703-t003:** Bond dissociation enthalpy (BDE) scale (XO → X + ^3^O) and (reverse) transfer thermodynamic reactivity scale (−TTRS: XO → X + ½ O_2_) (simply the reverse of the scale in Table 19 of reference [14]). All values are in kcal/mol. See Appendix A for Kekulé Structures.

XO (Oxide of Amine, Amide or Other)	BDE (XO → X + ^3^O)			−TTRS (XO → X + ½ O_2_)		
	Exp’t	B3LYP/6-31G*	M06/6-311G+(d,p)	Exp’t	B3LYP/6-31G*	M06/6-311G+(d,p)
CO_2_ (Carbon dioxide)	127.3	129.5	134.9	67.7	67.8	77.8
NO_2_ (Nitrogen dioxide)	73.5	75.9	76.0	13.9	14.2	18.9
1-Aza-1,3-cyclohexadiene N-oxide ^a^	-	69.2	68.7	-	7.5	11.6
1-Azacyclohexene N-oxide ^a^	-	65.9	66.0	-	4.2	8.9
2-Carboxylpyridine N-oxide	65.9	64.3	62.0	6.3	2.6	4.9
4-Cyanopyridine N-oxide	63.5	63.1	62.1	3.9	1.4	5.0
Pyridine N-oxide (PNO)	63.4	62.1	61.5	3.8	0.4	4.4
3-Cyanopyridine N-oxide	60.8	59.7	58.9	1.2	−2.0	1.8
1-Azabicyclo[2.2.2]octane N-oxide (Quinuclidine-N-oxide)	-	52.8	55.7	-	−8.9	−1.4
1-Azaadamantane-2-one N-oxide ^a^	-	48.8	50.3	-	−12.9	-6.8
Trimethylamine N-oxide (TMAO)	61.1 ^b^	48.6	51.5	1.5 ^b^	−13.1	−5.6
1-Azabicyclo[3.3.2]decane N-oxide ^a^	-	48.2	50.2	-	−13.5	−6.9
1-Azabicyclo[2.2.2]octan-2-one N-oxide ^a^ (2-Quinuclidinone N-oxide)	-	47.7	49.6	-	−14.1	−7.5
1-Azabicyclo[3.3.3]undecane N-oxide ^a^ (Manxine N-oxide)	-	46.3	47.2	-	−15.4	−9.9
1-Azabicyclo[4.3.3]dodecane N-oxide ^a^	-	39.3	40.4	-	−22.4	−16.7
1-Azabicyclo[3.3.1]nonan-2-one N-oxide ^a^	-	37.2	38.8	-	−24.6	−28.3
N-Methyl-2-pyrrolidinone N-oxide ^a^	-	34.6	35.8	-	−27.1	−21.3
1-Azabicyclo[3.3.3]undecane-2-one N-oxide ^a^ (2-Manxinone N-oxide)	-	29.9	31.4	-	−31.8	−25.7

^a^ Compound presently unknown. ^b^ See further discussion in text.

**Table 4 molecules-25-03703-t004:** Experimental and calculated N-O bond lengths (Angstroms) in pyridine N-oxide (PNO) and trimethylamine N-oxide (TMAO).

E. Diffraction		Microwave	X-Ray	6-31G* ^a^	B3LYP/6-31G* ^b^	M06/6-311G+(d,p) ^b^
PNO	1.290 ± 0.015 ^c^	1.278 ± 0.01 ^d^	1.330 ± 0.009 ^e^	1.275 ^b^	1.274	1.262
TMAO	1.379 ± 0.003 ^a^	-	1.388 ± 0.005 ^f^	1.370 ^a^	1.356	1.346

^a^ See Reference [18]. ^b^ This Study. ^c^ See Reference [30]. ^d^ See Reference [31]. ^e^ See Reference [32,33]. ^f^ See Reference [34].

**Table 5 molecules-25-03703-t005:** Experimental and calculated proton affinities (PA, kcal/mol) of some amines and amine oxides investigated in this study. These data are from a compendium ^a^: values in kJ/mol are divided by 4.184 kJ/kcal to provide data in kcal/mol.

Amine or Amine Oxide	PA (Exp’t)	PA (B3LYP/6-31G*)	PA [M06/6-311G+(d,p)]
Pyridine N-oxide (PNO)	220.7	220.3	216.4
Pyridine	222.3	224.4	219.3
Trimethylamine	226.8	226.4	220.4
Piperidine	228.0	230.0	224.0
1-Azabicyclo[3.3.3]undecane	233.9	234.0	228.0
1-Azabicyclo[2.2.2]octane	235.0	235.1	229.2
Trimethylamine N-Oxide (TMAO)	235.0	238.1	231.1
1-Azabicyclo[3.3.2]decane ^b^	-	234.7	228.5
1-Azabicyclo[4.3.3]dodecane ^b^	-	238.9	221.4

^a.^See Reference [37]. ^b^.Compounds presently unknown.

**Table 6 molecules-25-03703-t006:** Calculated enthalpies (kcal/mol) of oxidation of selected amines or lactams by hydrogen peroxide (H_2_O_2_ → H_2_O) or dimethyldioxirane (DMDO → Acetone).

Amine or Lactam	H_2_O_2_ ^a^		DMDO ^b^	
	B3LYP/6-31G*	M06/6-311G+(d,p)	B3LYP/6-31G*	M06/6-311G+(d,p)
Pyridine ^c^	−24.2	−31.6	−30.6	−39.7
Trimethylamine ^d^	−10.7	−21.6	−17.1	−29.7
Quinuclidine	−14.9	−25.8	−21.3	−33.9
Manxine	−8.4	−17.3	−14.8	−25.4
2-Quinuclidinone	−9.8	−19.7	−16.2	−27.8
1-Aza-2-adamantanone	−10.9	−20.4	−17.3	−28.5
2-Manxinone	+8.0	−1.5	+1.6	−9.6
1-Azabicyclo[3.3.1]-nonanone	+0.7	−8.9	−5.7	−17.0
*N*-Methyl-2-pyrrolidinone	+3.3	−5.9	−3.1	−14.0

^a^ It is useful to compare experimental [ΔH°**_f_** (g)] values and calculated values for ground-state H_2_O_2_ (−32.5 kcal/mol), H_2_ (0.0 kcal/mol) and ^1^Δ_+g_ O_2_ (22.7 kcal/mol).[21] The experimental enthalpy difference (−55.2 kcal/mol) could be compared with the B3LYP/6-31G* value (−56.6 kcal/mol) and the M06/6-311G+(d,p) value (−68.4 kcal/mol). The latter technique appears to provide an enthalpy for H_2_O_2_ that was 13 kcal/mol too high and, as such, appeared to overestimate the exothermicities of H_2_O_2_ oxidations by this quantity. ^b^ Calculated values for ΔH°**_f_** (g) for dimethyldioxirane and its isomer were −25.3 kcal/mol and −98.8 kcal/mol, respectively (B3LYP/6-31G* [40]) Corresponding values obtained by Etim, E.E.; Arunan, E. See Reference [41]: dimethyldioxirane: G3, −26.1 kcal/mol; G4MP2, −27.7 kcal/mol; G4, −27.7 kcal/mol; Corresponding values for methyl acetate: G3, −91.7 kcal/mol; G4MP2, −95.1 kcal/mol; and G4, −95.1 kcal/mol. In the present work, the arithmetic means of −26.7 kcal/mol and −95.1 kcal/mol (−98.8 kcal/mol, exp’t) were employed. Comparison of the average for DMDO with the accurate experimental value for methyl acetate yielded an enthalpy difference of −72.1 kcal/mol favoring the ester. The corresponding calculated enthalpy differences were −73.6 kcal/mol (B3LYP/6-31G*) and −83.6 kcal/mol [M06/6-311G+(d,p)]. The latter technique appeared to provide an enthalpy for DMDO that was 10 kcal/mol too high and, as such, appeared to overestimate exothermicities of dimethyldioxirane oxidations by this quantity. ^c^ Employing ΔH°**_f_** (g) = +29.8 kcal/mol for pyridine N-oxide yielded ΔH_r_ = −29.1 kcal/mol for oxidation by H_2_O_2_ and –29.0 kcal/mol for oxidation by DMDO (see footnotes a and b above). ^d^ Employing ΔH°**_f_** (g) = −7.2 kcal/mol for trimethylamine N-oxide yielded ΔH_r_ = −26.8 kcal/mol for oxidation by H_2_O_2_ and −26.7 kcal/mol for oxidation by DMDO (see footnotes a and b above).

**Table 7 molecules-25-03703-t007:** Comparisons between experimental and computational reaction enthalpies (kcal/mol) for the hydrogenation reactions depicted in Equations (7)–(9).

Reduction Reactions	ΔH_1_	ΔH_2_	ΔH_3_	ΔH_1_ + ΔH_2_ + ΔH_3_
Pyridine (Equation (7))				
Experimental	+2.7	−26.8	−20.8	−44.9
B3LYP/6-31G*	+7.2	−26.9	−18.3	−38.0
M06/6-311G+(d,p)	+3.5	−29.0	−23.1	−48.6
Benzene (Equation (8))				
Experimental	+5.7	−26.6	−28.3	−49.2
B3LYP/6-31G*	+9.2	−26.9	−28.6	−46.3
M06/6-311G+(d,p)	+5.2	−28.9	−30.6	−54.3
Pyridine N-oxide (Equation (9))				
Experimental	n/a	n/a	n/a	n/a
B3LYP/6-31G*	+0.1	−23.6	−3.5	−27.0
M06/6-311G+(d,p)	+3.6	−26.4	−8.2	−31.0

**Table 8 molecules-25-03703-t008:** Experimental and calculated values (kcal/mol) for saturation with 3 mol of H_2_.

Molecule	Exp’t	B3LYP/6-31G*	M06/6-311G+(d,p)	B3PW/6-31G*	B3PW/6-31G**	B3PW/6-311G+(d,p)
Benzene	−49.2	−46.3	−54.3	−53.8	−52.8	−48.2
Phenol	−45.4	−45.5	−54.3	−52.4	−51.3	−48.5
Phenoxide ^a^	−20.2	−12.9	−24.7	−19.1	−18.3	−17.6
Pyridine	−44.9	−38.0	−48.6	−45.5	−45.8	−43.1
Pyridine-N-oxide	n/a	−24.0	−38.2	−30.6	−30.5	−31.7

^a^ The ΔH°**_f_** (g) for cyclohexanoxide was estimated by comparison of the values [21] for isopropoxide, isopropanol, and cyclohexanol. The experimental uncertainties [21] listed for phenoxide and isopropoxide were ±10 kJ/mol (ca ± 2.4 kcal/mol). Therefore, the uncertainty for cyclohexanoxide might be as high as ±3 kcal/mol, which combined with the uncertainty for phenoxide yielded an experimental value for the saturation of phenoxide of −20 ± 5 kcal/mol, compatible with all calculational values in this table except for B3LYP/6-31G*.

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
