# Peer review of "Computational Study of Selected Amine and Lactam N-Oxides Including Comparisons of N-O Bond Dissociation Enthalpies with Those of Pyridine N-Oxides"

_molecules, 2020, doi:10.3390/molecules25163703_

Round 1

Reviewer 1 Report

Estimation of accurate bond dissociation energies (BDEs) is a challenging task. The values for N-oxides are especially difficult to measure experimentally due to its molecular property originated from Zwitterionic N-O bonds. In this work, Liebman, Greenberg and co-worker tabulated DFT-based N-O bond BDEs for important molecular scaffold in organic chemistry. The authors thoroughly walked through previous experimental and computational attempts to measure the BDEs and highlighted their limitations. By employing widely employed combinations of functionals and basis sets, isodesmic reaction scheme enabled to reproduce experimental BDEs of a number of compounds. Important finding in this piece of work is the importance of resonance contributions to the N-O BDEs. The authors nicely showcased this with a representative comparison between pyridine N-oxide and trimethylamine N-oxide. Overall, this reviewer believe that Molecules is an appropriate forum for this work. Below are minor points that might strengthen the presentation.

  1. Chemical structures for the compounds in Table 3 need to be drawn as an independent figure.
  2. Higher level calculation of BDEs might further support the accuracy of the functional/basis set employed. This reviewer suggests performing benchmark study for at least few compounds by utilizing ab initio calculations such as the composite basis set method (CBS-QB3).
  3. Few typos in Section 2.5 need to be fixed.

Author Response

Response to Reviewer 1 Comments

Point 1: Chemical structures for the compounds in Table 3 need to be drawn as an independent figure.

Response 1: Chemical structures with names clarifying Table 3 are now in Supplementary Information

Point 2: Higher level calculation of BDEs might further support the accuracy of the functional/basis set employed. This reviewer suggests performing benchmark study for at least few compounds by utilizing ab initio calculations such as the composite basis set method (CBS-QB3).

Response 2: We did employ CBS-QB3 as suggested and applied it to the BDE values for pyridine N-oxide (PNO) and trimethylamine N-oxide (TMAO). The new value was very consistent with our previous calculated and experimental values for PNO. The new value for TMAO was certainly somewhat higher than our calculated value. But the new data support our conclusion (and prediction) that BDE of PNO is significantly higher than that of TMAO and that redetermination of the thermochemistry of TMAO is appropriate.

Point 3: Few typos in Section 2.5 need to be fixed.

Response 3: Fixed typos

Reviewer 2 Report

Manuscript ID molecules-887867 belongs to that class of classic articles in the framework of the methodological studies. Authors perform an enormous study in selected amines and lactam N-oxides, comparing with previous results already reported at a level of theory that would be inadequate today. The computational methodology employed by the Authors is, however, rather acceptable. A few typos should be amended: “cm-1” all as superscript at line 164, two parentheses at line 170, amongst others, but beyond these minor issues, I strongly recommend the use of Kekulé models to illustrate the amine and lactam structures. In this regard, I was totally unable to find the structure of, for instance, “manxine”, and I am completely sure that (in general) this would help readers to “follow the message”. In opinion of this referee, this article might be of interest for those part of the computational chemistry community interested in methodological studies (particularly in thermodynamics), although the paper itself does not report any finding of high relevance to the theoretical community.

Author Response

Point 1: Authors perform an enormous study in selected amines and lactam N-oxides, comparing with previous results already reported at a level of theory that would be inadequate today. The computational methodology employed by the Authors is, however, rather acceptable.

Response 1: Agree about level of calculations although satisfied that we checked BDE's of PNO and TMAO using CBS-QB3

Point 2: A few typos should be amended: “cm-1” all as superscript at line 164, two parentheses at line 170, amongst others

Response 2: Fixed typos

Point 3: I strongly recommend the use of Kekulé models to illustrate the amine and lactam structures. In this regard, I was totally unable to find the structure of, for instance, “manxine”, and I am completely sure that (in general) this would help readers to “follow the message”.

Response 3: Chemical structures with names clarifying Table 3 are now in Supplementary Information

Reviewer 3 Report

From a computational point of view, this work is fantastic.
A complete review of the existing values in the literature is made for N-oxide derivatives, followed by a computational study and a very careful critical analysis. I find it interesting to use the experimental values of NO molecules already studied to calibrate the computational procedure to estimate for molecules not known. The authors used more than one method and different basis set.
Here are some small suggestions:
1) Page 3, Table 1, the first row of the table is a little confusing. Please try to correct it.
2) I had a little difficulty in understanding how the authors obtained the values of Table 1. For the reader's ease, it might be better to explain how those values were obtained.
3) There is a clear difference between experimental and computational BDE values. What is the opinion of the authors in determining a correction parameter for computational values from empirical data?
4) Page 7, eq. (4) please indicate what each symbol in the equation means.

Author Response

Point 1: Page 3, Table 1, the first row of the table is a little confusing. Please try to correct it.

Response 1: Clarified Table 1 through better formatting.

Point 2: I had a little difficulty in understanding how the authors obtained the values of Table 1. For the reader's ease, it might be better to explain how those values were obtained.

Response 2: Clarified in Table 1 caption the relative values in each column of each of the three sets

Point 3: There is a clear difference between experimental and computational BDE values. What is the opinion of the authors in determining a correction parameter for computational values from empirical data?

Response 3: As to a correction factor- just not enough experimental thermochemical data.

Point 4: Page 7, eq. (4) please indicate what each symbol in the equation means.

Response 4: Clarified eqn 4